# Safety of paclitaxel-coated devices in the femoropopliteal arteries: A systematic review and meta-analysis

**Chenyang Zhang**, **Guosheng Yin** *

Department of Statistics and Actuarial Science, University of Hong Kong, Hong Kong, China

* gyin@hku.hk

## Abstract

### Background

Clinical benefit of paclitaxel-coated devices for patients with peripheral arterial disease has been confirmed in randomized controlled trials (RCTs). A meta-analysis published in 2018 identified late mortality risk over a long follow-up period due to use of paclitaxel-coated devices in the femoropopliteal arteries, which caused enormous controversy and debates globally. This study aims to further evaluate the safety of paclitaxel-coated devices by incorporating the most recently published data.

### Methods

We searched for candidate studies in PubMed (MEDLINE), Scopus, EMBASE (Ovid) online databases, government web archives and international cardiovascular conferences. Safety endpoints of interest included all-cause mortality rates at one, two and five years and the risk ratio (RR) was used as the summary measure. The primary analysis was performed using random-effects models to account for potential clinical heterogeneity.

### Findings

Thirty-nine RCTs including 9164 patients were identified. At one year, the random-effects model yielded a pooled RR of 1.06 (95% CI [0.87, 1.29]) indicating no difference in short-term all-cause deaths between the paclitaxel and control groups (crude mortality, 4.3%, 214/5025 versus 4.5%, 177/3965). Two-year mortality was reported in 26 RCTs with 382 deaths out of 3788 patients (10.1%) in the paclitaxel arm and 299 out of 2955 patients (10.1%) in the control arm and no association was found between increased risk of death and usage of paclitaxel-coated devices (RR 1.08, 95% CI [0.93, 1.25]). Eight RCTs recorded all-cause deaths up to five years and a pooled RR of 1.18 (95% CI [0.92, 1.51]) demonstrated no late mortality risk due to use of paclitaxel-coated devices (crude mortality, paclitaxel 18.2%, 247/1360 versus control 15.2%, 122/805).

**Data Availability Statement:** All relevant data are within the paper and its Supporting information files.

**Funding:** GY is funded by the Research Grants Council of Hong Kong (17308420). The funders

had no role in study design, data collection and analysis, decision to publish, or preparation of the manuscript.

**Competing interests:** The authors have declared that no competing interests exist.

## Conclusions

We found no significant difference in either short- or long-term all-cause mortalities between patients receiving paclitaxel-coated and uncoated devices. Further research on the longer-term safety of paclitaxel usage (e.g., 8- or 10-year) is warranted.

## Registration

PROSPERO, CRD42021246291.

## Introduction

Peripheral arterial disease (PAD) is a common cardiovascular disease which affects about 10% of the general population worldwide [1]. PAD is known as one of the leading causes of cardiovascular morbidity and mortality and the progressed PAD can result in severe impairment of functional capacity and deterioration of life quality [2] Standard percutaneous transluminal angioplasty (PTA) has been used as the first line endovascular therapy for the treatment of PAD but is associated with a high rate of vessel restenosis and limited durability in clinical efficacy [3]. In recent years, the newly developed drug-coated balloons (DCBs) and drug-eluting stents (DESs) using paclitaxel in the femoropopliteal arteries (FPAs) have shown substantial improvements in reducing restenosis, target lesion revascularization and late lumen loss [4–6].

However, the safety of long-term use of paclitaxel DCB and DES has raised great concerns. In December 2018, Katsanos et al. [7] demonstrated the association between the use of paclitaxel DCB or DES and increased long-term risk of all-cause deaths compared with PTA or bare metal stent (BMS) in their meta-analysis with 28 randomized controlled trials (RCTs). As a result, the U.S. Food and Drug Administration (FDA) issued warning letters to health care providers regarding potential risk of paclitaxel devices [8, 9] and a preliminary analysis conducted by FDA reported an approximately 50% increase in all-cause mortality for patients treated with paclitaxel-coated devices versus control [9].

The SWEDEPAD [10] and BASIL-3 [11] trials, which investigated paclitaxel-coated devices in patients with PAD, were both temporarily suspended in December 2018 due to concerns on patient safety. The BASIL-3 trial restarted patient enrollment in September 2019 according to recommendations from an independent expert advisory group [12]. An unplanned interim analysis of the SWEDEPAD trial [10] showed no difference in mortality between the paclitaxel-coated and uncoated groups at one year or during the entire follow-up period thus far, and based on these results, it was decided to resume enrollment in the SWEDEPAD trial in March 2020. Such conflicting evidence led to further controversy over risks and benefits of paclitaxel-used devices for the treatment of PAD [13].

We conducted this systematic review and meta-analysis to update findings from previous reports by including more recently published RCTs and explore short- and long-term safety issues of paclitaxel DCBs or DESs in the FPAs.

## Methods

This systematic review and meta-analysis were conducted in compliance with the Preferred Reporting Items for Systematic Reviews and Meta-Analyses (PRISMA) statement [14] and registered in the PROSPERO database (CRD42021246291; https://www.crd.york.ac.uk/prospero/).

## Study selection

We performed extensive online searches of the PubMed (MEDLINE), Scopus, EMBASE (Ovid) databases, government web archives (US Food and Drug Administration and European Medicines Agency) and oral presentations in international cardiovascular conferences for eligible studies from August 2018 to June 2022. Our analysis also included 28 RCTs investigated in Katasanos et al. [7], for which the literature search was up to August 2018. There were no restrictions on publication language or publication status. The detailed searching strategies were given in S1 Table.

We included studies which met the following inclusion criteria: (1) randomized controlled trial; (2) patients with peripheral arterial disease of the FPA; (3) head-to-head comparison between paclitaxel-coated/eluting balloons/stents and standard percutaneous transluminal angioplasty or bare metal stent; (4) follow-up period ≥1 year; (5) outcome measures of interest reported. The exclusion criteria were: (1) retrospective cohort study or non-randomized trial; (2) treatment in vessels other than FPA; (3) studies that compared paclitaxel-coated/eluting devices with other drug-coated/eluting devices.

## Data extraction and assessment of risk of bias

Assessment of eligible studies and data extraction were performed by two investigators separately and they resolved disagreements by discussion. Titles and abstracts (if available) of studies were reviewed and full texts of those meeting the inclusion criteria were further screened for eligibility. For each included trial, we collected information of the trial design, paclitaxel DCB and DES devices used in the intervention, baseline demographic characteristics and outcome measures of interest.

Two independent reviewers evaluated the quality of included RCTs using a revised Cochrane risk-of-bias tool for randomized trials (RoB 2.0) [15], which focuses on five domains of bias: bias arising from the randomization process; bias due to deviations from intended interventions; bias due to missing outcome data; bias in measurement of the outcome; and bias in selection of the reported results. Any discrepancies were resolved by discussion.

## Outcome measurement

In this meta-analysis, we focused on all-cause mortality at one year, two years and five years as safety endpoints to evaluate short- and long-term risks of paclitaxel-coated/eluting devices in the FPAs. If there were several studies (e.g., interim analysis) reporting the same trial, results extracted from the latest one were considered for quantitative analysis.

## Statistical analyses

The primary analysis investigated the number of all-cause deaths at several prespecified time points, for which the risk ratio (RR) with the corresponding 95% confidence interval (CI) was used as the summary measure. The pooled estimates were calculated by the random-effects model to account for heterogeneity due to differences in trial designs, use of paclitaxel devices and patient populations. The Mantel-Haenszel fixed-effects model and Bayesian meta-analysis under the binomial-logit framework were conducted as sensitivity analyses. The potential publication bias was visually evaluated by checking asymmetry of funnel plots [16] and statistically examined by Egger's test [17].

Subgroup analyses were conducted to assess the influence of paclitaxel dose levels and types of paclitaxel-coated devices on patient mortality rates. We also performed a meta-regression analysis under the Bayesian binomial-logit model using the proportion of patients with chronic

limb threatening ischemia (CLTI) in each arm as the covariate. Enrolled patients suffering from PAD consisted of those with CLTI and those with intermittent claudication (IC). Compared with IC, CLTI is an advanced stage of PAD with higher amputation and mortality rates [18] and might be one potential cause of heterogeneity among studies. All statistical analyses were performed using the R language version 4.0.3 (RStudio, Boston, MA) with the 'meta' package for frequentist meta-analysis and 'jagsUI' package for Bayesian meta-analysis and meta-regression.

## Results

The online search from databases and other sources identified 1384 publications based on the prespecified search strategy after deleting duplicate records, of which 1237 studies were excluded after screening titles and abstracts. Full texts of the remaining 147 articles/presentations were assessed for eligibility and 59 of them were qualified for inclusion in the systematic review and meta-analysis. The flow diagram of our study selection process is shown in Fig 1.

Overall, the selected 59 studies reported 39 unique RCTs including 9164 patients [6, 10, 19–75]. The design characteristics of eligible RCTs are provided in Table 1. Out of these 39 RCTs, 29 RCTs investigated the clinical effectiveness of paclitaxel DCB, four tested the paclitaxel-coated balloon in combination with a BMS and the other six were for DES. Almost all studies evaluated the performance of one single paclitaxel device at the nominal paclitaxel dose of 2.0 (7/39), 3.0 (21/39) and 3.5 $\mu$g/mm$^2$ (9/39) except the SWEDEPAD trial [10] in which multiple device brands were used at various dose levels. Thirty RCTs were conducted at multiple sites, three were two-center and six were single-center. Three RCTs were double-blinded to both patients and investigators for the treatment allocation, 24 were single-blinded to patients only and the other 12 were open-label studies.

There were three 3-arm RCTs (DEBATE-IN-SFA [38], ISAR-STATH [62] and THUNDER [24, 47]) and only patients receiving the paclitaxel DCB/DES and standard PTA/BMS were included in the meta-analysis, while observations from the BMS plus cilostazol (DEBATE-IN-SFA), directional atherectomy (ISAR-STATH) and PTA plus paclitaxel in the contrast medium (THUNDER) groups were removed. In the ZILVER-PTX trial, patients assigned to the PTA arm through the primary randomization underwent a secondary randomization to DES or BMS if PTA failed acutely [40, 59, 71] and we pooled the results of patients receiving DES during both randomizations in the analysis.

The average age was over 65 years in all studies and about 65% of patients were male. The number of patients with IC and CLTI at baseline based on the Rutherford classification was reported in 37 RCTs. In total, 2342 (29%) patients had CLTI and 5729 (71%) had IC. The majority of patients in the DEBATE-SFA [36], SWEDEPAD [10] and Ni et al. [52] trials had CLTI, and in 28 RCTs, IC only accounted for less than 20% of enrolled patients. The detailed demographic and angiographic characteristics at baseline of the included RCTs can be found in S2 and S3 Tables, respectively. All-cause mortality data at one, two and five years are shown in S4 Table.

All the included 39 RCTs were judged to be of high overall risk of bias and the main source of risk arose from the bias due to deviations from intended interventions. There existed noticeable visual difference between paclitaxel-coated/eluting devices and standard uncoated devices. Therefore, investigators were usually not blinded to treatment assignment, which might affect the clinical decision making during the follow-up period. Detailed risk of bias assessment for each included RCT is presented in S1 Fig.

### All-cause mortality

The one-year all-cause mortality was reported in all 39 RCTs with 8990 patients. The crude risk of death at one year was 4.3% (214/5025) in the paclitaxel arm and 4.5% (177/3965) in the

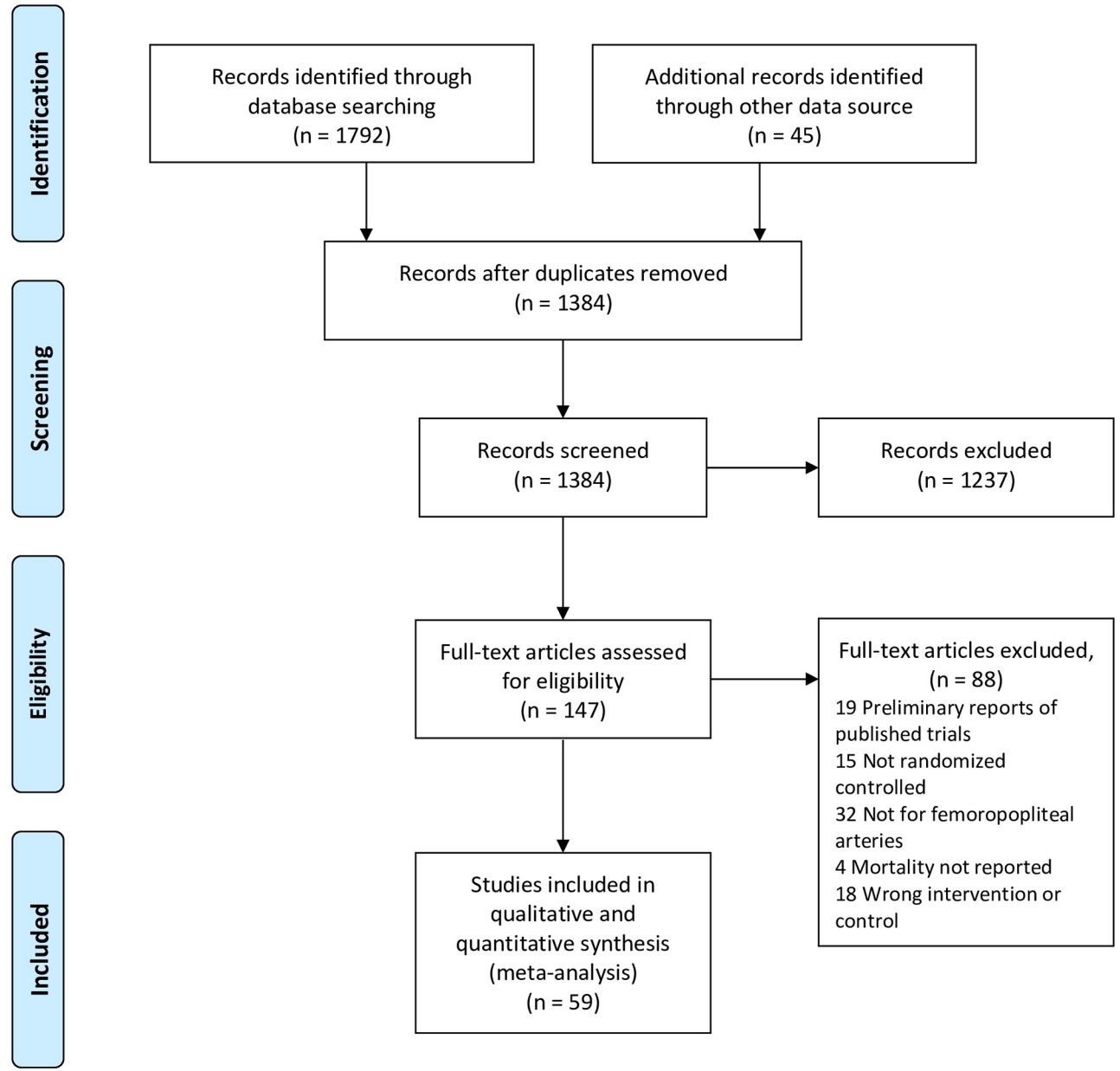

**Fig 1. Flow diagram of study selection.**

control arm. As shown in Fig 2, the random-effects model yielded a pooled RR of 1.06 (95% CI [0.87, 1.29]), suggesting no statistically significant difference in one-year mortality between the paclitaxel DCB/DES and control groups. There was no heterogeneity across the studies ($I^2$ = 0%; p = 0.98).

A total of 26 RCTs directly recorded all-cause deaths by two years. The unplanned interim analysis of SWEDEPAD trial [10] reported safety outcomes of paclitaxel-coated devices during a mean follow-up of 2.5 years. We estimated the number of deaths by two years in the SWEDEPAD trial [10] using the two-year cumulative incidence rate multiplied with the total number of patients in each group. Overall, at the two-year follow-up, 382 out of 3788 patients

**Table 1. Study characteristics of the 39 included randomized controlled trials in the meta-analysis.**

| Trial | Registration Number | Follow-up | Study Design | Location | Treatment | Control | Sample size, Intervention | Sample size, Control | Paclitaxel-Coated Device | Dose (µg/mm²) |
|---|---|---|---|---|---|---|---|---|---|---|
| ZILVER-PTX [40, 59, 71] | NCT00120406 | 5 | Multi-center; Open-label | Germany, Japan, USA | DES | PTA or BMS | 241 | 238 | ZILVER-PTX Stent by COOK Medical | 3 |
| THUNDER [24, 47] | NCT00156624 | 5 | Multi-center; Single-blind | Germany | DCB | PTA | 48 | 54 | Cotavance Balloon by Bavaria Medizin | 3 |
| IN.PACT SFA [6, 32, 39] | NCT01175850 / NCT01566461 | 5 | Multi-center; Single-blind | Germany | DCB | PTA | 220 | 111 | IN.PACT Admiral Balloon by Medtronic | 3.5 |
| FEMPAC [44] | NCT00472472 | 2 | Multi-center; Single-blind | Germany | DCB | PTA | 45 | 42 | Paccocath Balloon by Bavaria Medizin | 3 |
| LEVANT I [46] | NCT00930813 | 2 | Multi-center; Single-blind | Belgium, Germany | DCB | PTA | 49 | 52 | Lutonix Balloon by CR BARD | 2 |
| LEVANT II [26, 72] | NCT01412541 | 5 | Multi-center; Single-blind | Belgium, Germany, USA | DCB | PTA | 316 | 160 | Lutonix Balloon by CR BARD | 2 |
| ILLUMENATE EU [43, 49, 70] | NCT01858363 | 5 | Multi-center; Single-blind | Austria, Germany | DCB | PTA | 222 | 72 | Stellarex Balloon by Spectranetics | 2 |
| CONSEQUENT [23, 74] | NCT01970579 | 2 | Multi-center; Single-blind | Germany | DCB | PTA | 78 | 75 | SeQuent Please Balloon By B. Braun Melsungen AG | 3 |
| ISAR-STATH [62] | NCT00986752 | 2 | Two-center; Open-label | Germany | DCB +BMS | PTA +BMS | 48 | 52 | IN.PACT Admiral Balloon by Medtronic | 3.5 |
| ISAR-PEBIS [45] | NCT01083394 | 2 | Two-center; Open-label | Germany | DCB | PTA | 36 | 34 | IN.PACT Admiral Balloon by Medtronic | 3.5 |
| IN.PACT SFA JAPAN [33, 34] | NCT01947478 | 2 | Multi-center; Single-blind | Japan | DCB | PTA | 68 | 32 | IN.PACT Admiral Balloon by Medtronic | 3.5 |
| ACOART I [22, 29, 48] | NCT01850056 | 5 | Multi-center; Single-blind | China | DCB | PTA | 100 | 100 | Orchid Balloon by Acotec Scientific | 3 |
| FINN-PTX [37] | NCT01450722 | 2 | Multi-center; Open-label | Finland | DES | PTFE Bypass Graft | 23 | 18 | ZILVER-PTX Stent by COOK Medical | 3 |
| BATTLE [61] | NCT02004951 | 2 | Multi-center; Open-label | France | DES | BMS | 86 | 85 | ZILVER-PTX Stent by COOK Medical | 3 |

*(Continued)*

**Table 1.** (Continued)

| Trial | Registration Number | Follow-up | Study Design | Location | Treatment | Control | Sample size, Intervention | Sample size, Control | Paclitaxel-Coated Device | Dose ($\mu g$/ $mm^2$) |
|---|---|---|---|---|---|---|---|---|---|---|
| DEBATE-IN-SFA [38] | UMIN000010071 | 1 | Multi-center; Open-label | Japan | DES | BMS | 85 | 85 | ZILVER-PTX Stent by COOK Medical | 3 |
| DEBELLUM [27] | NA | 1 | Single-center; Open-label | Italy | DCB | PTA | 25 | 25 | IN.PACT Admiral Balloon by Medtronic | 3.5 |
| PACIFIER [57, 75] | NCT01083030 | 2 | Multi-center; Single-blind | Germany | DCB | PTA | 44 | 47 | IN.PACT Pacific Balloon by Medtronic | 3.5 |
| FAIR [31] | NCT01305070 | 1 | Multi-center; Single-blind | Germany | DCB | PTA | 62 | 57 | IN.PACT Admiral Balloon by Medtronic | 3.5 |
| BIOLUX P-I [60] | NCT01056120 | 1 | Multi-center; Single-blind | Austria, Belgium,France, Germany, Ireland,Israel, Latvia, Netherlands, Spain, Switzerland | DCB | PTA | 30 | 30 | Passeo-18 Lux Balloon by Biotronik | 3 |
| RANGER SFA [21, 69] | NCT02013193 | 1 | Multi-center; Single-blind | Austria,France, Germany | DCB | PTA | 71 | 34 | Ranger Balloon by Boston Scientific | 2 |
| ILLUMENATE pivotal [43, 54, 68] | NCT01858428 NCT01912937 | 5 | Multi-center; Single-blind | USA | DCB | PTA | 200 | 100 | Stellarex Balloon by Spectranetics | 2 |
| DEBATE-SFA [36] | NCT01556542 | 1 | Single-center; Open-label | Italy | DCB +BMS | PTA +BMS | 53 | 51 | IN.PACT Admiral Balloon by Medtronic | 3.5 |
| LEVANT JAPAN [50, 66] | NCT01816412 | 2 | Multi-center; Single-blind | Japan | DCB | PTA | 71 | 38 | Lutonix Balloon by CR BARD | 2 |
| RAPID [28, 67] | ISRCTN47846578 | 2 | Multi-center; Double-blind | Netherlands | DCB +BMS | PTA +BMS | 80 | 80 | Legflow Balloon by Cardionovum | 3 |
| EFFPAC [30, 41, 53] | NCT02540018 | 2 | Multi-center; Single-blind | Germany | DCB | PTA | 85 | 86 | Luminor-35 Balloon by iVascular | 3 |
| PACUBA [58] | NCT01247402 | 1 | Two-center; Single-blind | Austria | DCB | PTA | 35 | 39 | FREEWAY Balloon by Eurocor | 3 |
| FREEWAY [64] | NCT01960647 | 1 | Multi-center; Single-blind | Austria, Germany | DCB +BMS | PTA +BMS | 105 | 99 | FREEWAY Balloon by Eurocor | 3 |

*(Continued)*

**Table 1.** (Continued)

| Trial | Registration Number | Follow-up | Study Design | Location | Treatment | Control | Sample size, Intervention | Sample size, Control | Paclitaxel-Coated Device | Dose ($\mu g/mm^2$) |
|---|---|---|---|---|---|---|---|---|---|---|
| DRECOREST [35] | NCT03023098 | 1 | Single-center; Double-blind | Finland | DCB | PTA | 29 | 28 | IN.PACT Balloon by Medtronic | 3.5 |
| SWEDEPAD [10] | NCT02051088 | 2.49 (Mean, Ongoing) | Multi-center; Open-label | Sweden | DCB | PTA | 1149 | 1140 | Multiple device bands | NA |
| Falkowski et al. [25] | NA | 3 | Single-center; NA | Poland | DES | BMS | 126 | 130 | Zilver PTX Stent by Cook Medical | 3 |
| COPA CABANA [56] | NCT01594684 | 2 | Multi-center; Double-blind | Germany | DCB | PTA | 47 | 41 | Cotavance Balloon by MEDRAD | 3 |
| Liao et al. [63] | ChiCTR1800017055 | 1 | Single-center; Single-blind | China | DCB | PTA | 38 | 36 | Orchid Balloon by Acotec Scientific | 3 |
| RANGER II SFA [19, 20] | NCT03064126 | 1 | Multi-center; Single-blind | Austria, Belgium, Canada, Japan, New Zealand, USA | DCB | PTA | 278 | 98 | Ranger Balloon by Boston Scientific | 2 |
| BIOPAC [51] | NCT02145065 | 3 | Multi-center; Single-blind | Poland | DCB | PTA | 33 | 33 | Microcrystalline PAK Balloon by Balton Sp. z o.o., Warszawa, Poland | 3 |
| Ni et al. [52] | NCT03844724 | 1 | Multi-center; Single-blind | China | DCB | PTA | 93 | 99 | ZENFlow Balloon by Zylox Medical Device Inc | 3 |
| ORCHID CHINA [55] | ChiCTR1900023619 | 1 | Single-center; Single-blind | China | DCB | PTA | 30 | 30 | Orchid Balloon by Acotec Scientific | 3 |
| Ye et al. [65] | NA | 2 | Multi-center; Open-label | China | DCB | PTA | 100 | 100 | Reewarm™ PTX by Endovastec Co., Ltd | 3 |
| FREEWAY-CHINA [73] | NA | 1 | Multi-center; Open-label | China | DCB | PTA | 155 | 154 | FREEWAY Balloon by Eurocor | 3 |
| EMINENT [42] | NCT02921230 | 1 | Multi-center; Single-blind | Austria, Belgium, France, Germany, Ireland, Italy, Netherlands, Spain, Switzerland, UK | DES | BMS | 508 | 267 | ELUVIA Stent by Boston Scientific | 0.167 |

BMS: bare metal stent; DCB: drug-coated balloon; DES: drug-eluting stent; PTA: percutaneous transluminal angioplasty

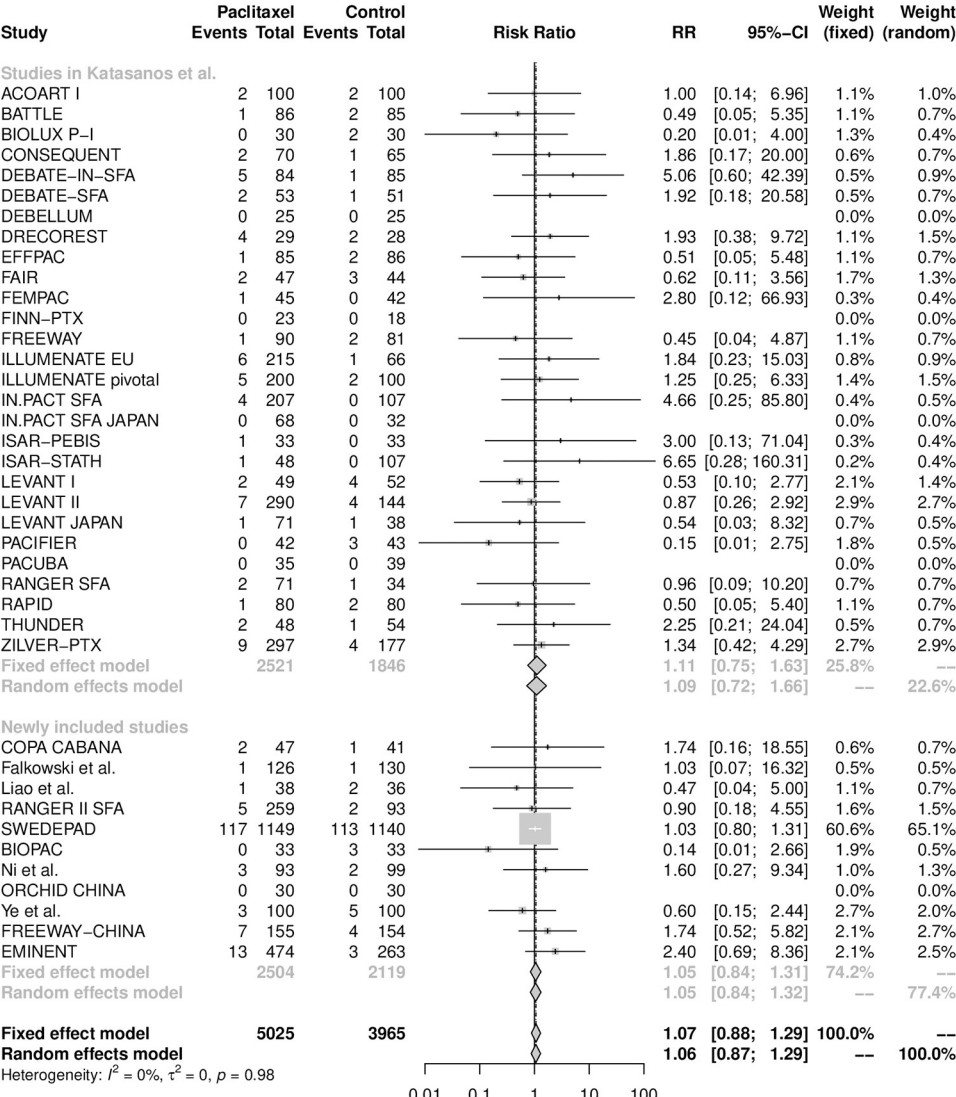

**Fig 2. Forest plot of all-cause mortality at one year.** RR is the risk ratio and CI represents the 95% confidence interval.

(10.1%) in the paclitaxel arm and 299 out of 2955 patients (10.1%) in the control arm had died. The forest plot in Fig 3 indicates that the use of paclitaxel-coated/eluting devices in the FPAs tended to increase the risk of death at two years but the evidence was not statistically significant (the random-effects model yielded RR = 1.08, 95% CI [0.93, 1.25]). We observed no heterogeneity among these 26 RCTs ($I^2$ = 0%; p = 0.47).

The five-year all-cause mortality was available only from eight RCTs with 2165 patients. The crude all-cause mortality at five years was 18.2% (247/1360) for patients receiving paclitaxel-coated/eluting devices and 15.2% (122/805) for those in the control group. The pooled RR calculated from the random-effects model was 1.18 (95% CI [0.92, 1.51]) (Fig 4). Although there was a 3.0% difference in the crude five-year mortality rate (95% CI [-0.2%, 6.2%]) between the paclitaxel-coated/eluting devices and control groups, the meta-analysis suggested that use of paclitaxel-coated balloons and stents did not significantly increase the risk of death

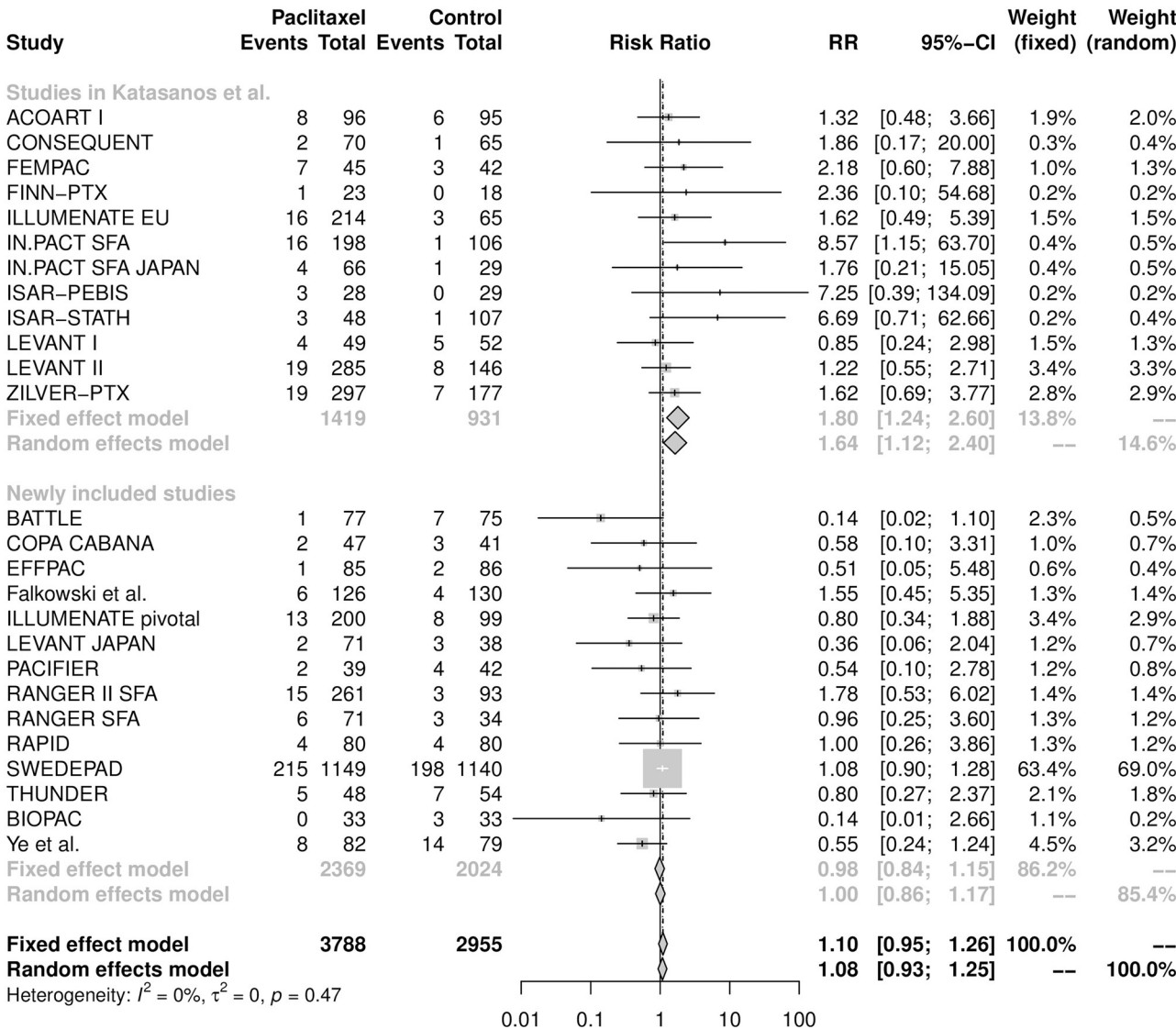

**Fig 3. Forest plot of all-cause mortality at two years.** RR is the risk ratio and CI represents the 95% confidence interval.

during the five-year follow-up period. No heterogeneity was found among these eight RCTs ($I^2$ = 28%; p = 0.20).

## Subgroup and sensitivity analysis

Subgroup analyses were performed to verify the influence of different paclitaxel interventions (DCB, DES or DCB+BMS) and dose levels (2.0, 3.0, 3.5$\mu$g/mm$^2$ stents or balloons) on all-cause mortality. We only analyzed subgroups with more than three RCTs. In subgroup analysis concerning different types of paclitaxel interventions, patients treated with DES had relatively higher one-year mortality rates compared with the control and in all intervention subgroups there was no significant difference in the risk of deaths at one, two and five years between the paclitaxel and control arms (Table 2). With respect to subgroup analysis of paclitaxel doses, studies using 3.5$\mu$g/mm$^2$ DCB showed a higher RR (the random-effects model yielded RR = 2.77, 95% CI [0.84, 9.22]) of two-year all-cause deaths compared to those using

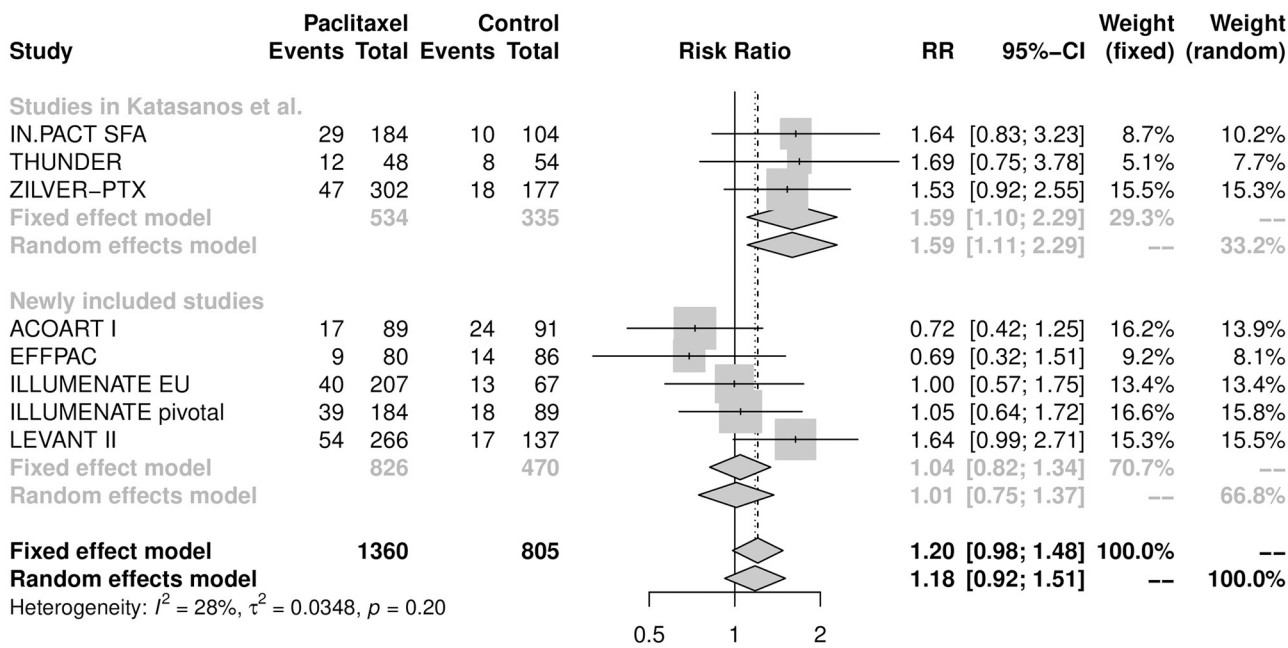

**Fig 4. Forest plot of all-cause mortality at five years.** RR is the risk ratio and CI represents the 95% confidence interval.

lower dose balloons and stents, while for all dose levels, no statistically significant result was found under the random-effects model.

We conducted sensitivity analyses using the fixed-effects model, various continuity correction methods for rare events and the Bayesian binomial-logit model on arm-level observations.

**Table 2. Subgroup analyses of all-cause mortality with subgroups including more than three randomized controlled trials.**

| | | Risk Ratio (95% CI) | |
|---|---|---|---|
| | Period | Random-effects | Fixed-effects |
| Paclitaxel intervention | | | |
| DCB | 1 year | 1.02 (0.83, 1.26) | 1.01 (0.82, 1.24) |
| DCB | 2 years | 1.06 (0.91, 1.23) | 1.08 (0.93, 1.26) |
| DCB | 5 years | 1.12 (0.86, 1.48) | 1.15 (0.92, 1.43) |
| DES | 1 years | 1.71 (0.83, 3.51) | 1.79 (0.89, 3.59) |
| DES | 2 years | 1.09 (0.41, 2.94) | 1.11 (0.61, 2.01) |
| DCB+BMS | 1 year | 1.07 (0.30, 3.76) | 1.10 (0.36, 3.35) |
| Dose level | | | |
| $2.0\mu g/mm^2$ DCB | 1 year | 0.90 (0.47, 1.74) | 0.92 (0.48, 1.75) |
| $2.0\mu g/mm^2$ DCB | 2 years | 1.05 (0.69, 1.58) | 1.07 (0.71, 1.61) |
| $3.0\mu g/mm^2$ DES | 1 year | 1.43 (0.59, 3.48) | 1.53 (0.66, 3.54) |
| $3.0\mu g/mm^2$ DES | 2 years | 1.09 (0.41, 2.94) | 1.11 (0.61, 2.01) |
| $3.0\mu g/mm^2$ DCB | 1 year | 0.94 (0.55, 1.63) | 0.89 (0.54, 1.48) |
| $3.0\mu g/mm^2$ DCB | 2 years | 0.86 (0.56, 1.33) | 0.85 (0.56, 1.29) |
| $3.5\mu g/mm^2$ DCB | 1 year | 1.42 (0.60, 3.40) | 1.35 (0.63, 2.90) |
| $3.5\mu g/mm^2$ DCB | 2 years | 2.77 (0.84, 9.22) | 3.05 (1.33, 7.01) |

BMS: bare metal stent; CI: confidence interval; DCB: drug-coated balloon; DES: drug-eluting stent.

We also considered exclusion of SWEDEPAD trial in the two-year meta-analysis because the data were estimated from the survival curves. As shown in S5 Table, in all cases the pooled estimate was close to that in the primary analysis and there was still no statistically significant increase of all-cause mortality due to the use of paclitaxel.

In addition, we conducted cumulative meta-analysis by years of publications on the two- and five-year all-cause mortality as a post-hoc analysis and the estimates from the random-effects model were shown in Fig 5 and S6 Table. For each follow-up period, the sequential plot started from the year in which ≥3 RCTs reported the endpoint of interest. As shown in Fig 3, for the 2-year all-cause mortality, studies in Katasanos et al. [7] reported an RR of 1.64 (95% CI [1.12, 2.40]) while the newly added studies yielded an RR of 1.00 (95% CI [0.86; 1.17]), which delivered contradictory results. The difference in the two-year all-cause mortality was statistically significant only in 2018 (RR = 1.51, 95% CI [1.05, 2.17] from 12 RCTs), and the two-year RR had been falling since 2017. Four RCTs (IN.PACT SFA [6], LEVANT II [26], THUNDER [24] and ZILVER-PTX [40]) reported five-year all-cause deaths by 2019, which yielded a pooled RR of 1.61 (95% CI [1.20, 2.16]). However, the pooled results turned to be insignificant after inclusion of four newly published RCTs (ACOART I [48], ILLUMENATE EU, ILLUMENATE pivotal [43] and EFFPAC [53]) in 2021 and 2022.

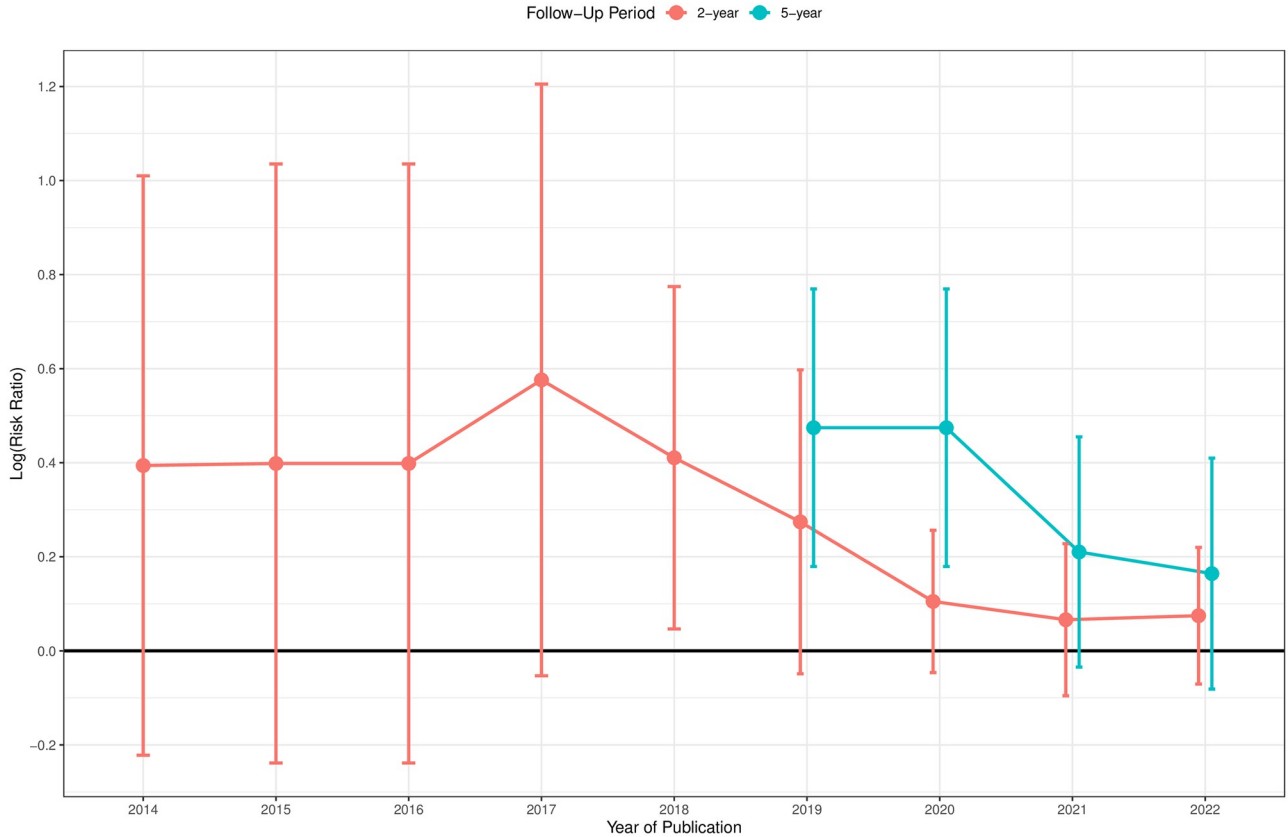

**Fig 5. Cumulative pooled treatment effect estimates (the logarithm of risk ratio) from the random-effects model by years of publications.**

### Publication bias

Visual inspection of funnel plots (S2 Fig) suggested no publication bias at one, two and five years due to approximately symmetrical shapes, and Egger's test produced confirmative results (one-year, p = 0.87; two-year, p = 0.82; five-year, p = 0.89), see S7 Table.

## Discussion

This systematic review and meta-analysis evaluated safety of the use of paclitaxel-coated/eluting devices in the FPAs and showed no significant difference in either short- or long-term all-cause mortality between patients receiving paclitaxel DCBs/DESs and those receiving the control (one-year RR, 1.06, 95% CI [0.87, 1.29]; two-year RR 1.08, 95% CI [0.93, 1.25]; five-year RR 1.18, 95% CI [0.92, 1.51]). Our findings are contrary to the results in Katsanos et al. [7] published in 2018, which reported significant mortality signals at two and five years due to paclitaxel usage. Compared with previous meta-analyses, our study included more recently published RCTs after 2020 and we can further investigate the trend of the increased risk of death due to paclitaxel over the publication time. The cumulative meta-analysis by years of publications (Fig 5) demonstrated that the inclusion of the most recently published studies reduced the difference in mortality between the paclitaxel and control arms and thus led to insignificant pooled results. For several recent studies, e.g., BATTLE (2-year) [61], COPA CABANA (2-year) [56], EFFPAC (2-year, 5-year) [30, 53], ACOART I (5-year) [48], patients in the control arm even had a higher risk of death compared with those receiving paclitaxel interventions. Pooled estimates from the meta-analysis are consistent across various statistical methods in the sensitivity analysis. The subgroup of the high paclitaxel dose ($3.5\mu g/mm^2$) showed a potential tendency towards a higher risk of death at one and two years compared with the low-dose one. No publication bias existed at any of the investigated follow-up periods.

The controversial meta-analysis of Katsanos et al. [7] has provoked heated discussions and debates worldwide on the long-term safety of paclitaxel-delivery devices in treating PAD. Subsequently, multiple studies were performed based on different data sources and statistical models, which repeatedly identified the late mortality signal. As a result, the US FDA released three warning letters and also conducted a preliminary analysis for the FDA-approved paclitaxel-coated devices, which confirmed the finding of an increased mortality rate due to the use of paclitaxel up to five years (RR = 1.72, 95% CI [1.25, 2.38]), but no significant difference was found in the two-year mortality (RR = 1.31, 95% CI [0.76, 2.29]) [8, 9, 26]. A Bayesian meta-analysis [76] identified only a borderline difference in mortality between paclitaxel and control arms beyond two years with inconclusive evidence. To overcome the lack of access to the original patient-level data, Albrecht et al. [75] and the Vascular Interventional Advances (VIVA) group [77] performed individual-patient-data meta-analyses based on patient-level data of four and eight RCTs, respectively. Albrecht et al. [75] indicated no evidence of increased risk of death using paclitaxel interventions at two years. The VIVA group reported a hazard ratio of 1.38 (95% CI [1.06, 1.80]) during a median follow-up of four years [77]. Klumb et al. [5] conducted a meta-analysis which included studies before February 2019 to evaluate the influence of paclitaxel-coated balloon angioplasty for patients with FPA diseases and they found no evidence on the increased risk of the two-year all-cause mortality after DCB. Based on individual patient-level data from two single-arm trials and two RCTs, Schneider et al. [78] demonstrated the safety of the use of paclitaxel-coated balloons to treat FPA diseases. An updated meta-analysis conducted by Dinh et al. [79] identified 34 RCTs before December 2020 and reported no association between the increased risk of all-cause mortality and the implementation of paclitaxel devices. Mathlouthi et al. [80] performed a retrospective study on patients

receiving paclitaxel-eluting stents and claimed no difference in the 2-year all-cause mortality in the real-world setting. Nevertheless, they observed a significantly higher risk of death among patients who required longer stents and higher paclitaxel doses. In addition, several retrospective cohort studies using large real-world datasets [81–84] were conducted to investigate safety outcomes of paclitaxel-coated devices while the late mortality signal could not be detected in any of these studies.

In conclusion, we observed no association between increased risk of death and the use of paclitaxel while with a longer follow-up period, the pooled results yielded a higher estimate of RR for paclitaxel versus control. On the other hand, paclitaxel DCBs and DESs have shown superiority over the standard uncoated endovascular therapies in reducing restenosis, target lesion revascularization and improving the quality of life. Considering the trustworthy clinical effectiveness and potential long-term risk, the risk-benefit profiles of paclitaxel DCBs and DESs are still of uncertainty. In real clinical practice, it is suggested to assess whether the treatment benefit of using paclitaxel in balloons and stents could outweigh the risk of mortality for each individual patient [8, 9, 69, 81].

As an update of the work by Katsanos et al. [7], this study has several limitations. The entire analysis was performed on the aggregate-level data. In the presence of censoring, survival analysis on time-to-event observations would be more appropriate, while for most of the included RCTs we had no access to individual patient data and the summarized binary data were used instead. When counting the number of patients, we excluded patients who were lost to follow-up to alleviate potential bias. Without the original patient-level data, we were unable to investigate the relationship between patient characteristics and survival or explore undetected sources of clinical heterogeneity. Most of the included RCTs focused on the evaluation of benefits of paclitaxel-coated devices (e.g., patency, late lumen loss, restenosis) and mortality was not the primary endpoint used for the trial design.

In the cumulative meta-analysis by years of publications, we observed a decreasing trend of RR for the two- and five-year all-cause mortality, indicating that paclitaxel-coated/eluting devices implemented in recent years were less toxic to patients with PAD. The improvement of products using paclitaxel might make substantial contributions to the treatment of lesions in the femoropopliteal arteries. However, in this study we did not consider this issue since there were few studies investigating the efficacy and safety of a specific paclitaxel device and the corresponding subgroup analysis could not provide convincing evidence. We focused on the use of paclitaxel-coated/eluting devices in FPA and excluded RCTs for infrapopliteal artery diseases. The safety of paclitaxel exposure in below-the-knee arteries is also a controversial topic [85] and several meta-analyses have been conducted [86–88]. Another limitation of our study is that we only considered a single safety endpoint of all-cause mortality and did not examine the association between the use of paclitaxel and other safety outcomes, e.g., thrombosis, allergy and major amputation of the target limb.

## Conclusion

This systematic review and summary-level meta-analysis showed that the use of paclitaxel DCBs and DESs was not associated with the increased short- or long-term mortality. Concerning well-proven clinical effectiveness of paclitaxel devices in the femoropopliteal arteries, further investigations including more RCTs with longer follow-up periods and individual patient-level data are warranted to shed more light on the risk-benefit profiles of paclitaxel usage in PAD patients.

## Supporting information

**S1 Fig. Risk of bias assessment.**
(TIF)

**S2 Fig. Funnel plots at (a) one year; (b) two years; (c) five years.**
(TIF)

**S1 Table. Search strategy of online databases.**
(DOCX)

**S2 Table. Detailed demographic statistics of included randomized controlled trials.**
(DOCX)

**S3 Table. Detailed angiographic statistics of included randomized controlled trials.**
(DOCX)

**S4 Table. All-cause mortality at one, two and five years of included randomized controlled trials.**
(DOCX)

**S5 Table. Sensitivity analyses of all-cause mortality.** Risk ratio (95% CI) for frequentist methods and odds ratio (95% equal-tailed CrI) for Bayesian methods.
(DOCX)

**S6 Table. Cumulative meta-analysis by year of publication.**
(DOCX)

**S7 Table. Egger's test of publication bias.**
(DOCX)

**S1 Checklist.**
(DOCX)

## Acknowledgments

We thank the two referees, the Associate Editor and Editor for their many constructive and insightful comments that have led to significant improvements in this paper.

## Author Contributions

**Conceptualization:** Chenyang Zhang, Guosheng Yin.

**Data curation:** Chenyang Zhang, Guosheng Yin.

**Formal analysis:** Chenyang Zhang, Guosheng Yin.

**Funding acquisition:** Guosheng Yin.

**Investigation:** Chenyang Zhang, Guosheng Yin.

**Methodology:** Chenyang Zhang, Guosheng Yin.

**Project administration:** Guosheng Yin.

**Resources:** Chenyang Zhang, Guosheng Yin.

**Software:** Chenyang Zhang.

**Supervision:** Guosheng Yin.

**Validation:** Chenyang Zhang, Guosheng Yin.

**Visualization:** Chenyang Zhang.

**Writing – original draft:** Chenyang Zhang, Guosheng Yin.

**Writing – review & editing:** Chenyang Zhang, Guosheng Yin.

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
