## [Decision Letter · Decision Letter 0]

21 Jun 2022

PONE-D-21-22189

Safety of paclitaxel-coated devices in the femoropopliteal arteries: A systematic review and meta-analysis

PLOS ONE

Dear Dr. Yin,

Thank you for submitting your manuscript to PLOS ONE. After careful consideration, we feel that it has merit but does not fully meet PLOS ONE’s publication criteria as it currently stands. Therefore, we invite you to submit a revised version of the manuscript that addresses the points raised during the review process.

The reviewers raised a number of concerns with your manuscript, including the methodological approach and the rationale for the study in the context of existing work, and the presentation of your data. The reviewers' comments can be viewed in full, below.

We look forward to receiving your revised manuscript.

Kind regards,

Natasha McDonald, PhD

Associate Editor

PLOS ONE

Journal Requirements:

3. Please upload a new copy of Figure S1 as the detail is not clear. Please follow the link for more information: https://blogs.plos.org/plos/2019/06/looking-good-tips-for-creating-your-plos-figures-graphics/" https://blogs.plos.org/plos/2019/06/looking-good-tips-for-creating-your-plos-figures-graphics/

Reviewers' comments:

Reviewer's Responses to Questions

**Comments to the Author**

1. Is the manuscript technically sound, and do the data support the conclusions?

Reviewer #1: Yes

Reviewer #2: Yes

2. Has the statistical analysis been performed appropriately and rigorously? 

Reviewer #1: Yes

Reviewer #2: Yes

3. Have the authors made all data underlying the findings in their manuscript fully available?

Reviewer #1: Yes

Reviewer #2: Yes

4. Is the manuscript presented in an intelligible fashion and written in standard English?

Reviewer #1: Yes

Reviewer #2: Yes

5. Review Comments to the Author

Reviewer #1: I would like to congratulate authors on this important manuscript. It well written manuscript with crucial implications in PAD. I agree with the findings as paclitaxel coated has no impact on patient mortality.

However to further strengthen the manuscript I would recommend following.

1. Please provide the location data on each study including countries where patients were enrolled.

2. Please provide the data on patient demographic and clinical characteristics if available.

3. As prior meta analysis has shown similar results, should include in more detail the novelty of this study ( mathlouthi et al journal of vascular surgery feb 2021, P Schneider et al May 2109 JACC)

4. Please include more limitations of the study.

Reviewer #2: The artile is technically good, but I'm affraid that the results are expect after the publication of the SWEDPAD study. Therefore, as it is written, the study adds little to the current evidence. However, it could be improved if the authors organized the way they present the results.

This study is not novel but it can help to understand what change in the risk estimation between 2018 (Katsanos) and 2022. Therefore, I suggest the authors that properly identify in their forest plots the new studies which have been published after 2018 so we identify them and understand their impact in the analysis. As you recognized in the discussion, being this an update of Katsanos, readers need to easily understand the differences between the works.

I saw you assessed the methodological quality of the studies using RoB 2 but you did not mention the results of the analysis in the results section. I suggest you do it.

"This systematic review and summary-level meta-analysis showed that the use of

paclitaxel DCBs and DESs was not associated with increased short- or long-term

mortality. However, at two and five years, we still observed a higher risk of death for

patients receiving paclitaxel-coated/eluting devices and the meta-analysis yielded

marginal results yet reaching statistical significance." I dont agree with this conclusion. You demonstrated that no increased risk was found. You have wrote a discussion based on the absence of a statistically significant risk of death. But you contradict yourself regarding the risk of death in the conclusion section. I think this is not in line with your findings. so I suggest you to simplify the conclusion section based on your actual findings.

6. PLOS authors have the option to publish the peer review history of their article (what does this mean?). If published, this will include your full peer review and any attached files.

Reviewer #1: **Yes: **Sawan Jalnapurkar

Reviewer #2: No

---

## [Author Response · Author response to Decision Letter 0]

17 Jul 2022

Responses to Comments from Reviewer 1 on Manuscript PONE-D-21-22189

We thank you for your careful review of our manuscript and many insightful comments. We have updated this meta-analysis and revised our manuscript by taking into account all of your comments, which led to a much better exposition of our work. Our point-by-point responses to major comments are given below, with your original comments copied in Italics for your convenience.

Comments:

1. Please provide the location data on each study including countries where patients were enrolled.

Response: Thanks for your suggestion. We have added the locations of the included randomized controlled trials (RCTs) in the revised Table 1.

2. Please provide the data on patient demographic and clinical characteristics if available.

Response: Thank you for this comment. We have added data on patient baseline demographic and angiographic characteristics of the included RCTs in S2 and S3 Tables (Supporting Information)

3. As prior meta analysis has shown similar results, should include in more detail the novelty of this study ( mathlouthi et al journal of vascular surgery feb 2021, P Schneider et al May 2109 JACC)

Response: Thank you for pointing out this issue. We added discussions on the comparisons between our meta-analysis and previous studies in the revised manuscript. The added text is quoted below.

Compared with previous meta-analyses, our study included more recently published RCTs after 2020 and we can further investigate the trend of the increased risk of death due to paclitaxel over the publication time.

……

Klumb et al. conducted a meta-analysis which included studies before February 2019 to evaluate the influence of paclitaxel-coated balloon angioplasty for patients with FPA diseases and they found no evidence on the increased risk of two-year all-cause mortality after DCB. Based on individual patient-level data from two single-arm trials and two RCTs, Schneider et al.demonstrated the safety of the use of paclitaxel-coated balloons to treat FPA diseases. An updated meta-analysis conducted by Dinh et al. identified 34 RCTs before December 2020 and reported no association between the increased risk of all-cause mortality and the implementation of paclitaxel devices. Mathlouthi et al. performed a retrospective study on patients receiving paclitaxel-eluting stents and claimed no difference in the 2-year all-cause mortality in the real-world setting. Nevertheless, they observed a significantly higher risk of death among patients who required longer stents and higher paclitaxel doses.

4. Please include more limitations of the study.

Response: Thank you for this suggestion. We discussed more limitations of our study in the Discussion section as follows.

In the cumulative meta-analysis by years of publications, we observed a decreasing trend of RR for the two- and five-year all-cause mortality, indicating that paclitaxel-coated/eluting devices implemented in recent years were less toxic to patients with PAD. The improvement of products using paclitaxel might make substantial contributions to the treatment of lesions in the femoropopliteal arteries. However, in this study we did not consider this issue since there were few studies investigating the efficacy and safety of a specific paclitaxel device and the corresponding subgroup analysis could not provide convincing evidence. We focused on the use of paclitaxel-coated/eluting devices in FPA and excluded RCTs for infrapopliteal artery diseases. The safety of paclitaxel exposure in below-the-knee arteries is also a controversial topic and several meta-analyses have been conducted. Another limitation of our study is that we only considered a single safety endpoint of all-cause mortality and did not examine the association between the use of paclitaxel and other safety issues, e.g., thrombosis, allergy and major amputation of the target limb.

Responses to Comments from Reviewer 2 on Manuscript PONE-D-21-22189

We thank you for your careful review of our manuscript and many insightful comments. We have updated this meta-analysis and revised our manuscript by taking into account all of your comments, which led to a much better exposition of our work. Our point-by-point responses to major comments are given below, with your original comments copied in Italics for your convenience.

Comments:

1. The article is technically good, but I'm afraid that the results are expect after the publication of the SWEDPAD study. Therefore, as it is written, the study adds little to the current evidence.

Response: Thanks for your comment. The SWEDEPAD trial enrolled 2289 patients and made an impactful contribution to the meta-analysis due to its large sample size. We agree that after the completion of the SWEDEPAD trial, the results of our meta-analysis might change. However, given the interim analysis of the SWEDEPAD trial, researches claimed that the results ‘did not show a difference between the groups in the incidence of death during 1 to 4 years of follow-up’, which is consistent with our conclusions. Given such an important controversial issue, our study gives a more comprehensive update of meta-analysis based on cumulative evidence. For this revision, ten new studies were added to our meta-analysis.

2. This study is not novel but it can help to understand what change in the risk estimation between 2018 (Katsanos) and 2022. Therefore, I suggest the authors that properly identify in their forest plots the new studies which have been published after 2018 so we identify them and understand their impact in the analysis. As you recognized in the discussion, being this an update of Katsanos, readers need to easily understand the differences between the works.

Response: Thanks for your suggestion. In Figs 2-4, we split all eligible RCTs into two subgroups: studies in Katsanos et al. and newly added studies. The revised forest plots show that for the 2-year all-cause mortality, studies in Katasanos et al. reported an RR of 1.64 (95% CI [1.12, 2.40]) while the newly added studies have an RR of 1.00 (95% CI [0.86; 1.17]), which delivers contradicted results. For the 5-year all-cause mortality , three studies in Katasanos et al. identified significantly increased risk of death due to the use of paclitaxel (RR=1.59, 95% CI [1.11, 2.29]]) while the five newly added studies demonstrated no association between the use of paclitaxel and increased risk of death (RR=1.01, 95% CI [0.75, 1.37]).

3. I saw you assessed the methodological quality of the studies using RoB 2 but you did not mention the results of the analysis in the results section. I suggest you do it.

Response: Thank you for this comment. We have added discussion on the evaluation of risk of bias as follows.

All the included 39 RCTs were judged to be of high overall risk of bias and the main source of risk arose from the bias due to deviations from intended interventions. There exists noticeable visual difference between paclitaxel-coated/eluting devices and standard uncoated devices. Therefore, investigators were usually not blinded to treatment assignment, which might affect the clinical decision making during the follow-up period. Detailed risk of bias assessment for each included RCT is presented in S1 Fig.

4. "This systematic review and summary-level meta-analysis showed that the use of paclitaxel DCBs and DESs was not associated with increased short- or long-term mortality. However, at two and five years, we still observed a higher risk of death for patients receiving paclitaxel-coated/eluting devices and the meta-analysis yielded marginal results yet reaching statistical significance." I dont agree with this conclusion. You demonstrated that no increased risk was found. You have wrote a discussion based on the absence of a statistically significant risk of death. But you contradict yourself regarding the risk of death in the conclusion section. I think this is not in line with your findings. so I suggest you to simplify the conclusion section based on your actual findings.

Response: Thank you for pointing out this issue. We have revised this part in the revised manuscript as follows.

However, at two and five years, we observed a potential tendency yet reaching statistical significance towards increased risk of death for patients receiving paclitaxel-coated/eluting devices. Concerning well-proven clinical effectiveness of paclitaxel devices in the femoropopliteal arteries, further investigations including more RCTs with longer follow-up periods and individual patient-level data are warranted to shed more light on the risk-benefit profiles of paclitaxel usage in PAD patients.

---

## [Decision Letter · Decision Letter 1]

2 Sep 2022

PONE-D-21-22189R1Safety of paclitaxel-coated devices in the femoropopliteal arteries: A systematic review and meta-analysisPLOS ONE

Dear Dr. Yin,

Thank you for submitting your manuscript to PLOS ONE. After careful consideration, we feel that it has merit but does not fully meet PLOS ONE’s publication criteria as it currently stands. Therefore, we invite you to submit a revised version of the manuscript that addresses the points raised during the review process.

We look forward to receiving your revised manuscript.

Kind regards,

Salvatore De Rosa

Academic Editor

PLOS ONE

Journal Requirements:

Additional Editor Comments:

In particular, carefully revise the tone of the discussion to accomodate the recommendations by Reviewer #3.

Reviewers' comments:

Reviewer's Responses to Questions

**Comments to the Author**

1. If the authors have adequately addressed your comments raised in a previous round of review and you feel that this manuscript is now acceptable for publication, you may indicate that here to bypass the “Comments to the Author” section, enter your conflict of interest statement in the “Confidential to Editor” section, and submit your "Accept" recommendation.

Reviewer #1: All comments have been addressed

Reviewer #3: (No Response)

2. Is the manuscript technically sound, and do the data support the conclusions?

Reviewer #1: Yes

Reviewer #3: Partly

3. Has the statistical analysis been performed appropriately and rigorously? 

Reviewer #1: Yes

Reviewer #3: I Don't Know

4. Have the authors made all data underlying the findings in their manuscript fully available?

Reviewer #1: Yes

Reviewer #3: Yes

5. Is the manuscript presented in an intelligible fashion and written in standard English?

Reviewer #1: Yes

Reviewer #3: Yes

6. Review Comments to the Author

Reviewer #1: (No Response)

Reviewer #3: I read with interest the manuscript # PONE-D-21-22189R1 by Dr Chenyang Zhang and Guosheng Yin entitled “Safety of paclitaxel-coated devices in the femoropopliteal arteries: A systematic review and meta-analysis”.

This is a systematic review and meta-analysis including 9164 patients from 39 RCTs evaluating the safety of paclitaxel-coated devices for the percutaneous treatment of femoro-popliteal atherosclerotic lesions.

The results indicate no difference in short-term all-cause deaths between the paclitaxel and control groups, but showing a potential tendency towards increased risk of death at two and five years.

The manuscript is well written and the topic is of interest although a growing body of evidence has been accumulated in the last few years.

The manuscript went through a first revision process and authors appropriately replied to the majority of the queries.

I suggest to re-consider the sentences were a trend towards increased risk of death is highlighted, since this is not strongly supported by the findings, especially when looking at more recently published RCTs (after 2020).

7. PLOS authors have the option to publish the peer review history of their article (what does this mean?). If published, this will include your full peer review and any attached files.

Reviewer #1: **Yes: **Sawan Jalnapurkar

Reviewer #3: No

---

## [Author Response · Author response to Decision Letter 1]

5 Sep 2022

We thank you for your careful review of our manuscript and many insightful comments. We have revised our manuscript by considering all of your comments, which led to a much better exposition of our work. Our point-by-point responses to major comments are given below, with your original comments copied in Italics for your convenience.

Comments:

To Editor:

1. In particular, carefully revise the tone of the discussion to accommodate the recommendations by Reviewer #3.

Response: Thanks for your suggestion. We have revised the mentioned sentences in the Abstract and Discussion sections following the recommendations by Reviewer #3.

To Reviewer #3

1. I suggest to re-consider the sentences were a trend towards increased risk of death is highlighted, since this is not strongly supported by the findings, especially when looking at more recently published RCTs (after 2020).

Response: Thank you for pointing out this problem. In the revised manuscript we have modified sentences involving ‘a trend towards increased risk of death’ in the Abstract and Discussion sections.

---

## [Editor Report · Decision Letter 2]

26 Sep 2022

Safety of paclitaxel-coated devices in the femoropopliteal arteries: A systematic review and meta-analysis

PONE-D-21-22189R2

Dear Dr. Yin,

We’re pleased to inform you that your manuscript has been judged scientifically suitable for publication and will be formally accepted for publication once it meets all outstanding technical requirements.

Kind regards,

Salvatore De Rosa

Academic Editor

PLOS ONE

Additional Editor Comments (optional):

The authors have amended the manuscript according to the criticism raised. I have no further comments
---

## [Editor Report · Acceptance letter]

2 Oct 2022

PONE-D-21-22189R2 

Safety of paclitaxel-coated devices in the femoropopliteal arteries: A systematic review and meta-analysis 

Dear Dr. Yin:

I'm pleased to inform you that your manuscript has been deemed suitable for publication in PLOS ONE. Congratulations! Your manuscript is now with our production department. 

Kind regards, 

on behalf of

Dr. Salvatore De Rosa 

Academic Editor

PLOS ONE